# Classifying atopic dermatitis: protocol for a systematic review of subtypes (phenotypes) and associated characteristics

Amy R Mulick,[1] Victoria Allen,[1] Hywel C Williams,[2] Douglas J C Grindlay,[2] Neil Pearce,[1,3] Katrina Abuabara,[4] Sinéad M Langan[1]

¹Department of Non-Communicable Disease Epidemiology, London School of Hygiene and Tropical Medicine, London, UK
²Centre of Evidence Based Dermatology, University of Nottingham, Nottingham, UK
³Department of Medical Statistics, London School of Hygiene and Tropical Medicine, London, UK
⁴Program for Clinical Research, Department of Dermatology, UCSF School of Medicine, University of California San Francisco, San Francisco, California, USA

**Correspondence to**
Amy R Mulick;
Amy.Mulick@lshtm.ac.uk

## ABSTRACT

**Introduction** Atopic dermatitis is a complex disease with differing clinical presentations. Many attempts have been made to identify uniform subtypes, or phenotypes, of atopic dermatitis in order to identify different aetiologies, improve diagnosis, estimate more accurate clinical prognoses, inform treatment and management or predict treatment efficacy and effectiveness. However, no consensus yet exists on exactly what defines these phenotypes or how many there are and whether they are genuine or statistical artefacts. This review aims to identify previously reported phenotypes of atopic dermatitis, the features used to define them and any characteristics or clinical outcomes significantly associated with them.

**Methods and analysis** We will search Ovid Embase, Ovid MEDLINE and Web of Science from inception to the latest available date at the time of the search for studies attempting to classify atopic dermatitis in humans using any cross-sectional or longitudinal epidemiological or interventional design. Primary outcomes are atopic dermatitis phenotypes, features used to define them and characteristics associated with them in subsequent analyses. A secondary outcome is the methodological approach used to derive them. Two reviewers will independently screen titles and abstracts for inclusion, extract data and assess study quality. We will present the results of this review descriptively and with frequencies where possible.

**Ethics and dissemination** Ethical approval is not required for this study as it is a systematic review. We will report results from this systematic review in a peer-reviewed journal. The main value of this study will be to inform further research.

**PROSPERO registration number** CRD42018087500

## Strengths and limitations of this study

► Prospectively registered review reported consistent with Preferred Reporting Items for Systematic Reviews and Meta-Analyses guidelines.
► This study will provide the first comprehensive review of current evidence on phenotypic subgroupings of atopic dermatitis. This is an important topic in a time of rapid development of new therapeutic options.
► It will be difficult to assess bias resulting from non-publication of studies or selective reporting of results.
► It may be difficult to synthesise the results, due to expected heterogeneity in identified phenotypes and in the outcomes/other characteristics explored.

## INTRODUCTION

Atopic dermatitis, also known as atopic eczema, eczema, neurodermatitis and Besnier's prurigo,[1 2] is a complex disease with variable clinical presentations, associations with other atopic diseases and clinical courses.[3] Traditionally, atopic dermatitis was characterised as an allergic disease of childhood, but it is now well-established that non-allergic forms exist and patients have been subdivided into those with and without atopy. However, there is much evidence suggesting that this dichotomisation is not clinically useful[4]; despite the use of names such as 'atopic dermatitis' or 'atopic eczema', up to two-thirds of patients with the disease are not atopic and atopy status does not predict outcomes or treatment responses.[5] Furthermore, genetic research in the last decade has led to a paradigm shift in understanding the aetiology of atopic dermatitis, from being considered primarily an allergic/immunological disorder to understanding the additional importance of skin barrier dysfunction.[6 7] Recent '-omics' research has further highlighted a role of lipid composition and immune pathways in disease pathology and phenotypic presentation.[8–10]

A phenotype is sometimes defined as a set of observable characteristics of an individual resulting from the interaction of its genotype with the environment.[11] This includes, for example, the individual's clinical characteristics, development and behaviour. In an epidemiological context, the word phenotype additionally refers to subtypes of a disease that are defined by different phenotypic

appearances. These subtypes may be referred to using a variety of terminology in addition to 'phenotypes': for example, they may be called disease classifications, subgroups, typologies, strata, patterns or taxonomies by different researchers. Subtypes or phenotypes are not to be confused with endotypes, which are subtypes of a disease defined specifically by different functional or pathobiological mechanisms. One endotype could give rise to multiple phenotypes.

Current atopic dermatitis phenotypes are based on clinical (exogenous) features, measurable genetic or immunological (endogenous) features, comorbidities or signs or symptoms course. In general, study participants have been characterised by a particular external characteristic, for example, history of eczema herpeticum (sometimes called an exophenotype) or internal characteristic, for example, elevated serum IgE, or low filaggrin protein expression or Th17 activation in skin (endophenotypes). Clinical comorbidities (such as asthma and hay fever or ichthyosis) and symptom trajectories have also been used to define phenotypes.[12–15] Recent studies have highlighted the heterogeneity of signs and symptoms trajectories and have suggested that different clinical courses are associated with immunological characteristics, disease locations and comorbidities such as food allergy.[16 17]

If there exist a number of atopic dermatitis phenotypes, each of which exhibits homogeneous disease characteristics, this has the potential to inform aetiology (ie, identify endotypes), patient prognosis and prediction of treatment efficacy. Finding simple, meaningful disease phenotypes can elucidate the different biological pathways underpinning them, leading to the discovery of endotypes (aetiology); knowledge of the markers for these pathways can assist clinicians in diagnosing and managing patients' symptoms more accurately, particularly on whether to expect persistence or remission (prognosis) and whether a potential treatment will work for them (prediction). At present, treatment for long-term atopic dermatitis is not based on disease phenotypes; treatment is symptomatic and may have associated toxicity.[18] The identification of meaningful phenotypes has potential to improve treatment strategies and provide biological insights into the future development of new ones.

No study has systematically reviewed the literature summarising the currently existing phenotypic classifications of atopic dermatitis. We will do this, aiming to identify previously reported phenotypes in studies specifically designed to identify subtypes of atopic dermatitis. We will also describe the features used to define them and whether they were predictive of or associated with any outcomes, concurrent conditions, treatment response or other relevant variables.

## METHODS AND ANALYSIS
### Eligibility criteria
We will search for studies with any cross-sectional or longitudinal epidemiological or interventional design whose primary or secondary aim is to define or identify subtypes/classifications/phenotypes of atopic dermatitis in humans of any age and gender. Features used to identify phenotypes could be based on either static or dynamic characteristics of their populations, and could include any feature of the disease, including clinical presentation, and genetic, immunological or molecular characteristics. We expect that most studies will contain only individuals with atopic dermatitis but some studies may have included individuals without it, for example, as a negative control group[15] or because a formal diagnosis was unavailable. We will include these studies. If included, the control population would be people known to be free of atopic dermatitis or who have a low probability of having it, including people who may have asthma, hay fever and other atopic diseases.

We will exclude: studies of localised eczema such as hand eczema and other types of eczema such as contact dermatitis and adult seborrheic dermatitis (for studies prior to 1990, we may include seborrheic dermatitis during infancy); literature reviews, books, book chapters, case reports, case series and in-progress phenotyping studies (abstracts), but not ongoing birth or other cohort studies; and conference proceedings and abstracts, as they are unlikely to provide sufficient detail on the definitions of atopic dermatitis phenotypes.

### Information sources
We will search Ovid Embase, Ovid MEDLINE and Web of Science from inception to the latest available date at the time of the search for publications in any language using English search terms. We will limit results to human studies published in original journal articles (published or in press, excluding retracted articles). We will interrogate the reference lists from the most recent two major review articles in each database.

### Search strategy
We will use conduct the following MEDLINE search:

((exp phenotype/ OR classification OR sub?type OR phenotyp* OR taxonomy OR disease type* OR disease typolog* OR stratif* OR strata) AND (exp dermatitis, atopic/ OR exp eczema/ OR exp neurodermatitis/ OR eczema* OR atopic dermatitis OR neurodermatitis OR besnier* prurigo))

Searches for the other databases will be matched as closely as possible to this using appropriate syntax and headings.

### Study records
#### Data management
Literature search results will be uploaded to Covidence web software, which will be used for all stages of the review process including title/abstract screening, full-text screening, data extraction, bias/quality assessment and process flow capture. We will develop and test screening questions based on the inclusion and exclusion criteria and a data extraction form based on the outcomes and pilot them on a subset of studies.

## Selection process

Two reviewers will scan all titles and abstracts independently. Publications that both reviewers record as 'not relevant' will not be retrieved for full-text review; the full text of all others will be retrieved into a 'short-list'.

Publications will be automatically included in this study if both reviewers independently assess them as meeting the inclusion criteria and excluded from this study where both assess them as not meeting the criteria. Disagreements will be resolved through referral to a third reviewer.

## Data collection process

During the short-list review, data from the texts in full will be extracted by two reviewers using a predesigned data extraction form and disagreements will be resolved by discussion among investigators.

## Data items

We will extract three domains of data from publications included in our study. In the event of subgroup analyses, such as studies reporting different atopic dermatitis phenotypes for men and women, we will extract data from the combined-group phenotypes if available. If unavailable, we will extract data for each phenotype separately. We believe it unlikely that more than one subgroup analysis will have been conducted in the absence of a main single analysis, but if this happens, we will extract data from the first subgroups reported in the Results section of the paper.

### Study data

We will extract the following items relating to the study: design, year(s) conducted, country/countries conducted in, setting conducted in (population-based, specialist etc), inclusion/exclusion criteria, number of participants, age range of participants, gender balance of participants, notable comorbidities (from study design) including, for example, proportions with other atopic diseases.

### Disease data

We will extract the following items relating to atopic dermatitis: definition codes/criteria, severity definition, prevalence and incidence (if relevant).

### Outcomes

We will extract the following items relating to atopic dermatitis phenotypes: a qualitative description of the phenotypes, features (variables) used to define phenotypes, age at the time of phenotype definition, proportion of individuals in each phenotype, qualitative description of any variables statistically significantly associated (in subsequent analyses) with the phenotypes (we will not report the effect estimates), statistical or other method used for classification, whether controls were used in classification algorithm.

## Outcomes and prioritisation

The outcomes for this review are atopic dermatitis phenotypes and associated characteristics, all of which are qualitative (ie, non-numeric) data.

Primary outcomes are: (1) atopic dermatitis phenotypes reported in published papers; (2) the features used to define the phenotypes (ie, the 'exposure' variables); and (3) the characteristics statistically significantly associated (in subsequent analyses) with the phenotypes, if any, which could include: long-term clinical outcomes such as disease persistence or severity; concurrent conditions such as other atopic disease; treatment response; genetic data (eg, *FLG* mutations); or any other variables reported in studies.

Our secondary outcome is a brief summary of the methodological approaches used to derive atopic dermatitis phenotypes.

We will not proactively seek particular values of the outcomes described above; we will collect any and all outcomes reported in the publications.

## Risk of bias

We will assess quality within and between studies using a checklist modified from the Grading of Recommendations Assessment, Development and Evaluation (GRADE) tool[19] for clinical trials or observational studies. Where relevant, instead of effect estimates and confidence intervals, we will assess equivalent parameters for the methods used to derive subtypes. We will additionally consider upgrading the final assessment by one level if the study is prospective, population-based or has been replicated.

### Within studies

We will treat the exposure as the feature(s) used to identify phenotypes, which will vary between studies, and the outcome as the identified phenotypes. We will give an initial rating of 'high quality' for randomised controlled trials or 'low quality' for observational studies and then judge risk of bias based on the extracted information on each of the domains relevant to the study type. Each item will be rated 'high risk of bias' or 'low risk of bias' and will be synthesised according to GRADE recommendations. Based on the synthesised judgement we will consider whether to downgrade or upgrade the initial quality assessment. Final possible quality assessments for individual studies are 'high', 'moderate', 'low' and 'very low'.

### Between studies

We anticipate the phenotype descriptions to vary between studies according to the features used to define them (exposure variables), so for a given outcome, it may not be possible to synthesise across studies. However, where it is possible, we will synthesise study results according to GRADE recommendations. Final possible quality assessments for the body of evidence are 'high', 'moderate' and 'low'.

## Data synthesis

The nature of the outcomes for this review precludes quantitative synthesis, so we will report our findings narratively. Our primary outcomes (atopic dermatitis phenotypes, the features used to define them and the outcomes/characteristics associated with them) are all

qualitative and we expect them to differ between studies, but where possible, we will group them into sensible categories and report frequencies.

For our secondary outcome, we will report the type and frequency of methodological approach used to derive the phenotypes.

We expect heterogeneity in all our outcomes because we have no reason to expect studies will have used similar protocols to explore phenotypes: for example, studies will probably have used different exposure variables to define phenotypes and will represent populations of people with different characteristics. To explore, this we will compare the: participant age and gender balance, study design, World Health Organization (WHO) region (Africa, Americas, Southeast Asia, Europe, Eastern Mediterranean and Western Pacific) and disease definition for each study, where possible. We will also look at hospital-based and population-based studies separately.

### Meta-biases

Outcomes in this review may be prone to meta-bias resulting from an absence of studies looking at important indicators of phenotypes, non-publication of study results or selective reporting of outcomes. For our qualitative outcomes, it will not be possible to measure this objectively. We will speculate on whether this is likely to be an important limiting factor in interpreting our results.

### Patient and public involvement

The research questions have been developed in consultation with Dr Sinéad Langan's Senior Clinical fellowship steering committee, which includes patient representation. The authors would like to thank Amanda Roberts for her contributions to discussions.

### ETHICS AND DISSEMINATION

This systematic review protocol was registered with the International Prospective Register of Systematic Reviews (PROSPERO) on 31 January 2018 and was last updated on 5 July 2018. Any amendments to the protocol will be documented on the PROSPERO site contemporaneously, with full explanation of any change. Ethical approval is not required for this study as it is a systematic review. We plan to submit a report of our findings for publication in a peer-reviewed journal. Findings from the report will also support the first author's PhD thesis.

**Contributors** ARM contributed to the design of the study, developed the search strategy, drafted the PROSPERO protocol and this manuscript and is the guarantor. VA contributed to the design of the study. HCW, NP and KA provided critical feedback on the PROSPERO protocol and drafts of the manuscript. DJCG approved the search strategy. SML contributed to the conception and design of the study, and provided critical feedback on the search strategy, PROSPERO protocol and drafts of the manuscript. All authors read, provided feedback and approved the final manuscript.

**Funding** This study was funded by a Wellcome Senior Clinical Fellowship to SML (205039/Z/16/Z). ARM and SML were supported by this grant. This research has received funding from the European Research Council under the European Union's Seventh Framework Programme (FP7/2007-2013)/ERC grant agreement no 668954.

**Competing interests** None declared.

**Patient consent** Not required.

**Provenance and peer review** Not commissioned; externally peer reviewed.

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
