## [Reviewer comments · BMJ Open]

ARTICLE DETAILS

TITLE (PROVISIONAL)	Classifying atopic dermatitis: protocol for a systematic review of subtypes (phenotypes) and associated characteristics
AUTHORS	Mulick, Amy; Allen, Victoria; Williams, HC; Grindlay, Douglas; Pearce, Neil; Abuabara, Katrina; Langan, Sinead

VERSION 1 – REVIEW

REVIEWER	Lin-feng Li Department of Dermatology Beijing Friendship Hospital, Capital Medical University 95 Yong-an Rd, Xicheng District, Beijing 100050, China
REVIEW RETURNED	16-Apr-2018

GENERAL COMMENTS	The project is well desinged. However, the eczema used here is actually atopic dermatitis. I suggest to use atopic dermatitis or atopic eczema to replace of eczema because eczema may include other types of dermatitis such as contact eczema.
--

REVIEWER	Matteo Megna Department of Dermatology, University of Naples Federico II
REVIEW RETURNED	01-Jun-2018

GENERAL COMMENTS	A well written and interesting study protocol to review available literature on eczema subtype classification and possible associated factors
---

REVIEWER	Kanwaljit K. Brar National Jewish Health, Denver, CO, United States
REVIEW RETURNED	15-Jun-2018

GENERAL COMMENTS	Thank you for your important protocol focusing on identifying and distinguishing isphenotypes of eczema. Our understanding of eczema has evolved beyond identifying skin barrier abnormalities in skin barrier proteins, such as filaggrin or loricrin, and newer literature is also examining variations in lipid composition (lipidomics), and immune pathways (Brunner et al JACI 2018). This may vary between age, and ethnicities; for example, in Asian patients, Th17 pathways may predominate resulting in a more psoriasiform phenotype (Noda et al JACI 2015). In evaluating secondary characteristics, I would encourage you to be inclusive of these newer distinguishing immunophenotypic features as they will likely play a role in therapeutic selection; or, alternatively, please specify in the protocol that focus will be on clinical and not molecular phenotypes. Additionally, consideration for associated diagnosis of food allergy should also be mentioned as this may also relate to atopic dermatitis phenotype, please see Roduit et al in JAMA
--

VERSION 1 – AUTHOR RESPONSE

Reviewer(s)' Comments to Author:

Reviewer: 1 Reviewer Name: Lin-feng Li Institution and Country: Department of Dermatology, Beijing Friendship Hospital, Capital Medical University, 95 Yong-an Rd, Xicheng District, Beijing 100050, China Please state any competing interests: none declared	
The project is well desinged. However, the eczema used here is actually atopic dermatitis. I suggest to use atopic dermatitis or atopic eczema to replace of eczema because eczema may include other types of dermatitis such as contact eczema.	We have changed all references to the disease under investigation from 'eczema' to 'atopic dermatitis', or rephrased the reference accordingly, including in the title. We have also referred to the World Allergy Association guidelines to explain that when we refer to atopic dermatitis, we include atopic eczema and eczema.
Reviewer: 2 Reviewer Name: Matteo Megna Institution and Country: Department of Dermatology, University of Naples Federico II Please state any competing interests: None to declare	
A well written and interesting study protocol to review available literature on eczema subtype classification and possible associated factors	Thank you; we think this will be an interesting and important review.
Reviewer: 3 Reviewer Name: Kanwaljit K. Brar Institution and Country: National Jewish Health, Denver, CO, United States Please state any competing interests: None declared	
Thank you for your important protocol focusing on identifying and distinguishing isphenotypes of eczema. Our understanding of eczema has evolved beyond identifying skin barrier abnormalities in skin barrier proteins, such as filaggrin or loricrin, and newer literature is also examining variations in lipid composition (lipidomics), and immune pathways (Brunner et al JACI 2018). This may vary between age, and ethnicities; for example, in Asian patients, Th17 pathways may predominate resulting in a more psoriasiform phenotype (Noda et al JACI 2015). In evaluating secondary characteristics, I would encourage you to be inclusive of these newer distinguishing immunophenotypic features as they will likely play a role in therapeutic selection; or, alternatively, please specify in the protocol that focus will be on clinical and not molecular phenotypes.	Thank you for highlighting some of the new developments in this field. We do intend to review phenotypes including both clinical and non-clinical facets of the disease, so to make this clearer we have added the following:  • 'Recent '-omics' research has further highlighted a role of lipid composition and immune pathways in disease pathology and phenotypic presentation' to the first paragraph of the introduction; • 'Th17 activation' to the third paragraph as an example of an endophenotype; • '... and could include any feature of the disease, including clinical presentation, and genetic, immunological or molecular characteristics, provided one of the primary or secondary aims was to define or identify subtypes/classifications/phenotypes of atopic dermatitis, to the Eligibility Criteria under Methods and Analysis.
Additionally, consideration for associated diagnosis of food allergy should also be mentioned as this may also relate to atopic	We will certainly collect this information if it is available in the results of relevant papers. We now mention this at the end of the third

dermatitis phenotype, please see Roduit et al in JAMA Pediatrics 2017.

paragraph in the introduction with the text 'and comorbidities such as food allergy' and appropriate citation.

Editor(s)' Comments to Author:

*FORMATTING AMENDMENTS (if any)
Required amendments will be listed here;
please include these changes in your revised
version:*

*- Patient and Public Involvement:
Authors must include a statement in the
methods section of the manuscript under the
sub-heading 'Patient and Public Involvement'.*

*This should provide a brief response to the
following questions:*

*How was the development of the research
question and outcome measures informed by
patients' priorities, experience, and
preferences?*

*How did you involve patients in the design of
this study?*

*Were patients involved in the recruitment to and
conduct of the study?*

*How will the results be disseminated to study
participants?*

*For randomised controlled trials, was the burden
of the intervention assessed by patients
themselves?*

*Patient advisers should also be thanked in the
contributorship statement/acknowledgements.*

*If patients and or public were not involved
please state this.*

We have added a paragraph at the end of the
'Methods and Results' section:

The research questions have been developed in
consultation with Dr Sinéad Langan's Senior
Clinical fellowship steering committee, which
includes patient representation. The authors
would like to thank Amanda Roberts for her
contributions to discussions.